# Design of Novel Tricaprylin-Incorporated Multi-Layered Liposomal System for Skin Delivery of Ascorbic Acid with Improved Chemical Stability

**DOI:** 10.3390/ph16010121

**Published:** 2023-01-13

**Authors:** Myoung Jin Ho, Dong Woo Park, Myung Joo Kang

**Affiliations:** College of Pharmacy, Dankook University, 119 Dandae-ro, Dongnam-gu, Cheonan 330-714, Chungnam, Republic of Korea

**Keywords:** ascorbic acid, lipo-oil-some, skin delivery, tricaprylin, photostability, topical application

## Abstract

L-ascorbic acid (Vit C) possesses a variety of dermatological functions in maintaining skin health and anti-aging properties. However, its topical application is challenging owing to its liability to light, oxygen, or heat. Therefore, in this study, a novel liposomal system, including a lipophilic neutral oil named a lipo-oil-some (LOS), was designed to improve the chemical stability and aid the skin absorption of Vit C. The vesicular systems were prepared using the ethanol injection method, employing phosphatidylcholine, cholesterol, dipalmitoyl-sn-glycerol-3-phosphoglycerol, and tricaprylin as neutral oil. The optimized LOS was characterized as follows: shape, multi-layered sphere; size, 981 nm; zeta potential, −58 mV; and Vit C encapsulation efficiency, 35%. The encapsulation of the labile compound into the novel system markedly enhanced photostability, providing over 10% higher Vit C remaining compared to Vit C solution or Vit C-loaded conventional liposome under a light intensity of 20,000 lx. On the other hand, the *ex vivo* skin permeation and accumulation of Vit C with the LOS system were comparable to those of smaller conventional liposomes (198 nm) in a Franz diffusion cell model mounted with porcine skin. Based on these findings, we concluded that the novel liposomal system could be utilized for skin delivery of Vit C with enhanced chemical stability.

## 1. Introduction

L-ascorbic acid (Vit C), a water-soluble vitamin, has been reported to have a variety of dermatological functions in maintaining skin health and anti-aging [1]. Vit C promotes collagen biosynthesis and differentiation of keratinocytes and reduces melanin synthesis and free radicals, which enhances skin immunity and antioxidant protection [2,3]. However, owing to the structural instability of the γ-lactone group, Vit C is easily degraded by light, aerobic conditions, high temperatures, alkalis, copper, and heavy metals [4,5]. In particular, as Vit C is highly sensitive to light, it is readily degraded to dehydroascorbic acid and 2,3-diketogulonic acid under UV irradiation conditions, losing its beneficial dermatological function [6]. Thus, the photosensitivity of Vit C remains a limitation when formulating pharmaceutical and cosmetic products [7]. Moreover, the dermatological application of Vit C, a hydrophilic molecule with a log*P* value of −1.85, is further hampered by low skin absorption. The main barrier for skin delivery is the stratum corneum (10–15 μm in thickness), which mainly comprises dead, flattened corneocytes, a tightly organized extracellular lipid matrix with a low moisture content of approximately 15–30%. This barrier function prevents most hydrophilic and large-molecular-weight drugs (>500 Da) from penetrating the intact skin layer [8].

To improve the chemical stability and skin absorption of water-soluble vitamins, different pharmaceutical approaches have been developed, including complexation [9,10], antioxidant combinations [11], and encapsulation in polymeric or lipidic carrier systems [12,13,14,15]. Liposomes are well-established spherical lipid vesicles that offer potential value in transdermal or skin drug delivery, as they are biodegradable, non-toxic, and can encapsulate both water-soluble and lipophilic substances [16]. The vesicular system consists of one or more phospholipid bilayers that enclose a discrete aqueous phase. Although the exact mechanisms for skin absorption are not yet fully understood, the proposed mechanisms of action involve (i) direct penetration of intact drug-laden vesicles into the skin layer, (ii) acting as permeation promotors through their lipid-softening property, (iii) “collision complex transfer” between the drug in the liposomal bilayer and stratum corneum, and (iv) facilitating skin absorption via appendageal pathways such as hair follicles and sweat ducts. Nevertheless, conventional liposomal carriers have a relatively loose bilayer membrane, allowing free passage of oxygen and light, which deteriorates the chemical stability of the loaded compounds. Actually, Vit C-loaded into phosphatidylcholine (PC)-based liposomes was degraded over 30% in 3 days even when stored under refrigerated, light-protected conditions [17]. Therefore, novel approaches that can further improve the chemical stability of labile compounds encapsulated in lipid vesicles are required.

As a novel approach for Vit C stabilization, we hypothesized that the incorporation of lipophilic neutral oil between the liposomal bilayers improves the stability of the labile compound by hampering the penetration of oxygen and light into the inner compartment. It has been reported that when tricaprylin, a type of medium-chain triglyceride, is included in a liposomal preparation, the lipophilic oil acts as a membrane stabilizer that tightens the adhesion of each vesicle, forming multi-vesicular liposomes [18,19]. Moreover, the excess oily component was located in the inner compartment in the form of droplets. Thus, we expected that lipophilic oils located between the bilayer or inner compartments would contribute to Vit C stabilization by forming a multi-layered physical barrier (Figure 1), along with its natural antioxidant effect [20]. Therefore, in this study, we designed a novel liposomal system including a lipophilic neutral oil, named a lipo-oil-some (LOS), to improve the chemical stability and skin absorption of Vit C. As neutral oil, tricaprylin, a water-immiscible oil with high lipophilicity (logP value of 7.41), was employed along with phospholipids, and cholesterol (Chol), to strengthen the physical barrier function of the liposomal carrier. 

The fabrication process and composition of the LOS system were optimized based on the vesicular size, homogeneity, and loading efficiency of Vit C. Vesicular morphology, in vitro release profile, and photostability of the optimized LOS system were evaluated in comparison with conventional liposome (CL). Moreover, the skin permeation and accumulation behaviors of Vit C-loaded LOS were evaluated using a Franz diffusion cell model mounted with porcine skin. 

## 2. Results and Discussion

### 2.1. Establishment of the Fabrication Process of the Vit C-Loaded LOS System

To prevent the degradation of labile compounds with improved skin delivery, a novel tricaprylin-incorporated liposomal system named LOS was designed. Phosphatidylcholine, a biocompatible phospholipid, was chosen as the principal component of the lipid bilayer, and cholesterol was combined with PC to increase the strength of the bilayer and control the release of Vit C molecules. A negatively charged phosphatidylglycerol, DPPG, was further incorporated to provide electrostatic repulsion between vesicles, preventing aggregation and precipitation of the colloidal system. Tricaprylin, a medium-chain triglyceride, was explored as a neutral, lipophilic oil for placement between the phospholipid bilayer or inside the vesicle. The concentrations of Vit C and lipid components in LOS were set to 20 mg/mL and 10 mg/mL, respectively. The LOS system was adjusted to pH 3.3–3.6, as Vit C was stable in succinate buffer-based weakly acidic conditions.

To fabricate the novel vesicular system of Vit C, an ethanol injection method, a well-established simple process with a high loading efficiency for hydrophilic materials, was employed [21]. Compared to the film hydration and freeze-drying methods, the ethanol injection method has the advantages of not only a simple, scalable manufacturing process but also the absence of harmful organic solvents such as chloroform and dichloromethane [22]. In the ethanol injection method, PC, DPPG, Chol, and tricaprylin were dissolved in ethanol and then added dropwise to an aqueous medium containing Vit C to form Vit C-encapsulated liposomal vesicles after evaporation of the organic solvent. Thereafter, the homogenizing process was performed to decrease the vesicular size with appropriate homogeneity. The temperature was set to 60 °C to effectively evaporate injected ethanol within 1 h. Vit C was stable during the LOS fabrication process, preserving over 90% of Vit C content (data not shown).

To provide a high Vit C encapsulation efficiency in the LOS system, different volume ratios of the water and oil phases during the LOS fabrication process were explored. The loading efficiency of substances in liposomes is an important parameter in the design of the fabrication process and composition. As shown in Figure 2, as the volume ratio of the oil phase to water phase decreases, the particle size tended to decrease; the vesicular sizes with the o/w ratios of 1:1, 1:2, and 1:2.5 were determined to be 840, 924, and 956 nm, respectively. As the o/w ratio decreases, merging and/or integration with adjacent vesicles during vesicle formation might be diminished with lowered lipid concentration, resulting in a decrease in vesicular size. In contrast, as the volume ratio of the oil phase to water phase decreased, the loading efficiency of Vit C gradually increased; the loading efficiencies with the o/w ratios of 1:1, 1:2, and 1:2.5 were determined to be 31.3, 31.5, and 38%, respectively. The low concentration of ethanol in the mixture might lower the solubility of phospholipids in the o/w mixture, inducing the rapid association of lipid molecules. This rapid vesicle formation of the vesicle facilitated the encapsulation of Vit C molecules into the inner compartment. Accordingly, the volume ratio of the oil phase and water phase was fixed at 1:2.5 to further fabricate LOS formulations.

### 2.2. Optimization of the Vit C-Loaded LOS System

Different amounts of Chol, DPPG, and tricaprylin were combined with PC to formulate a novel liposomal system, and the vesicular size, zeta potential, and Vit C encapsulation efficiency were evaluated. The concentration of PC in the LOS system was fixed at 20 mg/mL, and the amounts of DPPG, tricaprylin, and Chol were varied in the ranges of 0–1, 0–40, and 0–8 mg/mL, respectively (Table 1). First, the effect of the amount of

DPPG, a negatively charged phospholipid, on the physical properties and Vit C loading efficiency of LOS was evaluated. The negatively charged phospholipid was introduced into the LOS to induce electrostatic repulsion between the liposomal vesicles by enhancing the negative charge of the liposome membrane. The amount of DPPG did not significantly affect the particle size; the vesicle size of LOS1 (no DPPG addition), LOS2 (DPPG 0.2 mg/mL), LOS3 (DPPG 0.4 mg/mL), and LOS4 (DPPG 1 mg/mL) was measured in the range of 900–1200 nm (Figure 3A). As PC itself is anionic in weak acidic conditions, the surface charge of LOS1 prepared with no DPPG was measured to be −52.5 mV, and the surface charge decreased to −67.6 mV as the amount of DPPG was increased to 1 mg/mL (LOS4). As the amount of DPPG increased, the negative charge on the liposomal surface became stronger, and the loading of Vit C, a negatively charged molecule by the removal of enolic hydrogen ion, gradually decreased because of electrostatic repulsion. Load efficiencies (%) of Vit C in LOS1, LOS2, LOS3, and LOS4 were determined to be 38, 37, 35, and 32%, respectively. Taken together, the amount of DPPG in the LOS formulation was set to 0.4 mg/mL.

Next, the amount of tricaprylin, the key component in the LOS system, was varied in the range of 0–40 mg (CL, LOS5, LOS6, LOS7, and LOS8), and its effects on vesicular size, zeta-potential, and Vit C loading efficiency were evaluated. The vesicular size was 198 nm when neutral oil was not included in the composition (CL). Conversely, the incorporation of tricaprylin (LOS5–LOS8) markedly increased the vesicular size to between 900 and 1100 nm (Figure 3B). The incorporation of neutral oils between the bilayers might increase the vesicular size by increasing the attractive force between the vesicles via the London dispersion force. Previous studies have shown that oil (triglycerides) acts as an adhesive that connects individual vesicles to form large-sized multi-vesicular liposomes [18]. In contrast, the loading efficiency of Vit C in the vesicles varied markedly depending on the amount of tricaprylin. The loading efficiencies of Vit C in CL (without tricaprylin), LOS5 (4 mg/mL), LOS6 (10 mg/mL), LOS7 (20 mg/mL), and LOS8 (40 mg/mL) were determined to be 25, 29, 35, 23, and 17%, respectively. When tricaprylin was included below 10 mg/mL, the neutral oil increased the hydrophobicity of the liposomal membrane, thus preventing leakage of the hydrophilic compound into the external aqueous phase. However, when an excess amount of neutral oil (>20 mg/mL) was included, tricaprylin might elevate vesicular hydrophobicity and even cause the collision of the oil droplet with Vit C molecules during vesicular formation, interfering with the encapsulation of Vit C in the inner compartment. However, the amount of neutral oil did not affect the surface charge of the LOS system (−57.5–−54 mV). The concentration of tricaprylin in the LOS system was set to 10 mg/mL with a high Vit C loading efficiency.

Cholesterol, a waxy, lipophilic ingredient, is reported to modulate membrane fluidity, physically improve the stability of the bilayer, and decrease leakage of the loaded active compounds [23]. In addition, the incorporation of Chol can increase the dermal permeability of liposomes [24,25,26,27]. Therefore, the effects of the amount of Chol on LOS properties, vesicular size, zeta potential, and Vit C loading efficiency were evaluated. When the amount of Chol was varied between 0 and 2 mg/mL, the vesicular size ranged between 915 and 1004 nm, and as the amount of Chol increased to 4 mg/mL, the particle size increased to 1390 nm (Figure 3C). An increase in vesicular size with increasing Chol content is consistent with findings of previous reports; as the concentration of Chol increases, more cholesterol molecules will be distributed in the phospholipid bilayer, causing an increase in the liposome’s mean diameter. The loading amount of Vit C in the LOS system was markedly affected by the quantity of Chol; the loading efficiency increased significantly as the amount of Chol increased to 2 mg/mL. Loading efficiencies of LOS 9, 10, 11, and 12 formulations with Chol levels of 0, 0.4, 1, and 2 mg/mL were determined to be 8, 21, 26, and 35%. The incorporation of Chol in the LOS system imparted rigidity and hydrophobicity to the bilayer, decreasing the membrane permeability of the hydrophilic antioxidant considerably. Conversely, when the amount of Chol was increased to 4 mg/mL (LOS 13) and 8 mg/mL (LOS 14), the loading efficiencies decreased to 13 and 10%, respectively. The extraordinary incorporation of Chol is known to interfere with the close packing of the liposomal bilayer by contributing to an increase in membrane fluidity [28]. Thus, the reduction in rigidity by excess addition of Chol might decrease the encapsulation efficiency of Vit C. Based on these findings, the LOS12 formulation prepared with 0.4 mg/mL of DPPG, 10 mg/mL of tricaprylin, and 2 mg/mL of Chol was selected and further evaluated based on photostability, in vitro release profile, and skin absorption capacity.

### 2.3. Morphology of Vit C-Loaded CL and LOS System

The morphological features of Vit C-loaded LOS (LOS 12) were examined using cryo-TEM. The liposomal vesicles were immobilized at approximately −80 °C to prevent disruption or breakage of the liposomal vesicles during the pretreatment process. As shown in Figure 4A, the CLs fabricated using the ethanol injection method were spherical and unilamellar (left panel) with a diameter < 200 nm. Minorly, multi-lamellar vesicles with 3–4 bilayers were prepared (right panel). During the preparation process, large-sized unilamellar vesicles or multi-lamellar vesicles might be first formed and split into small unilamellar vesicles by the shearing force obtained from the additional homogenization process, providing a homogenous vesicular diameter below 200 nm. 

In contrast, LOS (LOS 12) showed a multi-lamellar or multi-vesicular vesicle structure, possessing markedly larger dimensions than those of CL (Figure 4B). Multi-vesicular vesicles containing two or three small nanovesicles per large vesicle were shaped with a mean diameter of approximately 1 μm. The vesicular size of the LOS system observed by cryo-TEM was consistent with that determined by the dynamic light scattering method (981 nm). Although the fabrication process was identical to that of CL, the inclusion of tricaprylin in the composition resulted in a marked difference in the liposomal structure. Lipophilic tricaprylin molecules might be incorporated between the PC-based bilayers during emulsification and ethanol evaporation. Actually, Ren et al. (2018) reported that the hydrophobic region inside lipid bilayers began to be occupied by a medium-chain triglyceride, expanding internal volume in the lipid bilayers [29]. Some fraction of oil droplets that were not incorporated into the vesicles might induce coalescence or binding with the adjacent vesicles through strong interactions with the lipid bilayers, forming multi-vesicular, multi-lamellar vesicles. Medium-chain triglycerides contained during liposomal preparation are reported to be located at the corners or edges where lipidic membranes meet and stabilize the membrane boundaries, forming multi-vesicular liposomes [19]. Luo et al. (2019) revealed that the incorporation of a medium-chain triglyceride into a liposomal bilayer caused the gel–liquid crystalline phase transition, increasing the fluidity of the phospholipid membrane [30]. The incorporation of neutral oils in lipid bilayers perturbates the van der Waals interaction between hydrophobic domains of adjacent phospholipid molecules, as well as disturbing the ordered alignment of lipid chains in the gel phase of phospholipid layers. Accordingly, when the weight ratio of medium-chain triglycerides to phospholipids was set below 1.75:1, a structure closer to liposomes rather than lipid emulsions was shaped, and when the weight ratio of medium-chain triglycerides to phospholipids was increased to 7.5:1, an emulsion structure was formed. As the amount of medium chain triglyceride increases (7.5:1), the lack of internal space in the liposomal bilayers for oil loading resulted in structural conversion into an oil-in-water type emulsion. In this study, the amount of tricaprylin applied to formulate the LOS system was equal to or less than that of phospholipids, so a stable liposomal bilayer structure was maintained, with no transformation into an oil-in-water emulsion. In addition, cholesterol contained in the LOS system contributed to the stabilization of the lipid bilayer. On the other hand, Vit C is a very hydrophilic anti-oxidant with a log*P* value of −1.85 and has known to be located in the inner compartment of CLs [31,32]. In designing the LOS system, tricaprylin is incorporated between lipid bilayer, but Vit C has poor solubility in tricaprylin (about 0.7 μg/mL). Therefore, Vit C molecules are located in the inner aqueous compartment of the LOS system, similar to CL. The enhancement of lipophilicity of liposomal vesicles with the formation of a multi-lamellar structure in the LOS system is expected to improve the chemical stability of internally located Vit C molecules.

### 2.4. Photostability of Vit C-Loaded CL and LOS System

The change in the Vit C content in the Vit C solution, Vit C-loaded CL, and LOS formulations under forced light exposure (light intensity of 20,000 lx) was evaluated. As previously reported, Vit C dissolved in aqueous media is principally degraded into dehydroascorbic acid and subsequently into 2,3-diketogulonic acid [33]. The irradiation of light in aqueous media resulted in the generation of reactive oxygen species [34], which accelerated the degradation of Vit C. For instance, under 4000 lx of fluorescent light exposure, the oxidation of Vit C in the microemulsion system gradually increased [35]. In addition, Vit C content (mg/L) in pasteurized milk stored under 1100, 2400, and 5800 lx of light irradiation was decreased from 12.93 mg/L to 8.38, 7.70, and 7.09 mg/L, respectively [36]. 

As speculated, the Vit C content in the Vit C solution, Vit C-loaded CL, and LOS gradually decreased in all formulations after 8 and 24 h of light exposure at an intensity of 20,000 lx (Figure 5). However, the decrease in Vit C content in the LOS system was markedly diminished compared with that in the Vit C solution and Vit C-loaded CL. After 8 h of light exposure, the decrease in Vit content in LOS was only 2%, whereas the decrease in Vit C solution and Vit C-loaded CL was measured to be approximately 7% and 15%, respectively. This trend was consistent with the Vit C stability data obtained after 24 h of light exposure, where the decrease in Vit C content in LOS was 12%, whereas that in the Vit C solution and Vit C-loaded CL was determined to be above 20% in both formulations. In the Vit C-loaded CL formulation, the photostability of Vit C was not improved compared to that of the Vit C solution because the unilamellar membrane could not effectively prevent the penetration of the irradiated light. Conversely, in the case of the LOS system, the neutral oil-incorporated lipophilic multi-lamellar membranes impeded the penetration of light more effectively, thus preventing the decomposition of the labile compound entrapped in the vesicles.

### 2.5. In Vitro Release Profile of Vit C from the Liposomal Systems

The in vitro release profiles of Vit C from the Vit C solution, Vit C-loaded CL, and LOS were evaluated using the Franz diffusion cell method (Figure 6). Vertical diffusion cells are commonly employed to assess the drug release and permeation of semi-solid external preparations. A regenerated cellulose membrane with a pore size of 8 kDa was placed between the donor and receptor compartments. The pore size (8 kDa) of the membrane was assumed to be sufficient to allow Vit C molecules to pass with no penetration of the liposomal vesicles.

In the case of the Vit C solution, the hydrophilic, small molecule was freely diffused out into the receptor compartment, showing over 95% release after 2 h. In contrast, both liposomal systems exhibited a biphasic pattern, showing an initial rapid release followed by retarded release. During the initial period (approximately 2 h), the free Vit C molecules present outside the liposomal vehicles rapidly migrated to the receptor compartment through diffusion. The release rates of CL and LOS after 2 h was determined to be 70% and 64%, respectively. Subsequently, Vit C molecules inside the liposome progressively moved through the lipid bilayer to the external phase and subsequently diffused into the receptor compartment, exhibiting a delayed release profile for 24 h. Although the LOS vesicle possessed a multi-layered membrane, the difference in release rate between LOS and CL was not noticeable. LOS exhibited only a 5–10% lower dissolution rate than CL from 2 to 12 h after the release test, suggesting that the small molecule could pass through the interstices of the lipid bilayer, regardless of the presence of the neutral oil droplets in the bilayer. Both liposomal systems were further explored to evaluate their *ex vivo* skin permeation and accumulation profiles with a controlled release profile.

### 2.6. Ex Vivo Skin Absorption of Vit C following Topical Application of Vit-C Loaded CL or LOS

The skin permeation and accumulation behaviors of the Vit C-loaded LOS system were comparatively evaluated with those of the Vit C-loaded CL system using a Franz diffusion cell model mounted on porcine dorsal skin (Figure 7). The permeability of several chemical compounds through porcine skin was found to be similar to that of human skin, with a comparable stratum corneum thickness of 21–26 μm, lamellar organization, hair follicle density, and other morphological aspects [37,38,39]. In a previous report, negatively-charged liposomes promoted skin accumulation of Vit C in the epidermis and dermis, providing a 7-fold higher permeation rate than the Vit C solution [14]. The permeation rate of Vit C from liposomal vehicles was found to be influenced by the lipid composition and physical characteristics of the vesicular system, such as the elasticity of the membrane or surface charge [40].

The skin permeation profile of Vit C after topical application of Vit C-loaded CL or LOS is illustrated in Figure 7A. Relevant permeation parameters, such as the amount of Vit C permeated (mg/cm^2^) after 24 h, flux (mg·cm^−2^·h^−1^), lag time (h), and permeability coefficient (10^−3^·cm/h) are presented in Table 2. Following topical administration of both liposomal formulations, Vit C, a small molecule, began to penetrate the skin instantaneously; the lag times of Vit C-loaded CL and LOS were determined to be 1.5 and 2.1 h, respectively, with no significant difference. The Vit C molecules presented outside of the lipid vesicle were rapidly released through the membrane in the in vitro release test; in the same manner, Vit C molecules present outside the liposome would begin to promptly penetrate the skin, with no difference between CL and LOS systems. Subsequently, the skin permeation profile of Vit C across the relevant layer reached a steady state in both liposomal systems, providing a proportional increase in the amount of Vit C that permeated for 24 h. The amount of Vit C permeation after 24 h was determined to be 0.16 and 0.2 mg/cm^2^ with the CL and LOS systems, respectively, which corresponds to 8% and 10% of the amount of Vit C initially loaded, respectively.

Although LOS has a larger vesicular size (982 nm) than CL (198 nm), it is disadvantageous for skin contact. In a skin absorption study in porcine skin, LOS displayed skin absorption of Vit C at a level comparable to that of CL. The permeability coefficient (10^−6^·cm/h) was comparable between the liposomal systems, with coefficient values of 0.718 and 0.909 (10^−6^·cm/h) for CL and LOS, respectively. Moreover, the amount of Vit C deposited after 24 h of LOS or CL application was comparable between LOS and CL. The amount of skin deposition after LOS or CL treatment was both 0.10 mg/cm^2^, which corresponds to 5.0% of the loading amount of Vit C in the donor compartment. These skin absorption data suggested that the novel LOS system acted effectively as a penetration enhancer, although it was challenging to penetrate the skin in an intact vesicle form because of its increased particle size. The skin penetration efficiency of liposomal vesicles was reported to depend not only on vesicular size but also on lipid composition, surface charge, flexibility and elasticity of vesicular membranes, and types of liposomes. Tricaprylin contained in the LOS system is lipophilic; however, skin-absorbable oil has been reported to possess a skin-softening effect. Accordingly, tricaprylin contained in LOS might act as a permeation promoter through its lipid-softening property or by promoting transfer between the drug in the liposomal bilayer and stratum corneum. Nevertheless, we are currently investigating the influence of lipid composition, surface charge, flexibility, and elasticity of vesicular membranes and their combination with hydrogel systems on the stability and skin absorption of Vit C to design a more sophisticated liposomal system.

## 3. Materials and Methods

### 3.1. Materials

Vit C powder (purity > 99.9%, *w/w*), cholesterol, tricaprylin (glyceryl trioctanoate), 1,2-dipalmitoyl-sn-glycerol-3-phosphoglycerol, succinic acid, formic acid, and sodium thiosulfate were purchased from Sigma-Aldrich (St. Louis, MO, USA). Phosphatidylcholine (PC) was obtained from Lipoid Korea (Geumcheon, Seoul, Republic of Korea). Absolute ethanol was purchased from Duksan Reagents (Ansan, Gyeonggi-do, Republic of Korea), and HPLC-grade acetonitrile was purchased from J.T. Baker (Phillipsburg, NJ, USA). All other chemicals were of analytical grade and were used without further refinement.

### 3.2. Fabrication of Vit C-Loaded Liposomal Formulations

Vit C-loaded liposomal formulations, including CL and LOSs (Table 1), were prepared using the ethanol injection method after a slight modification of the previously reported process [41,42] under light-shielded conditions. The aqueous phase was prepared by dissolving 25 mg of Vit C in 2 mL of 10 mM succinate buffer (pH 3.0) using a magnetic stirrer. The oil phase was prepared by dissolving 50 mg of PC, 0–20 mg of Chol, 0–2.5 mg of DPPG, and 0–100 mg of tricaprylin in 1 mL of ethanol at 50 °C using a magnetic stirrer. Thereafter, 1 mL of the oil phase solution was rapidly injected into 2 mL of the water phase at 450 rpm at 50 °C for 10 min. The o/w mixture was then homogenized at 15,000 rpm for 10 min using a high-speed homogenizer to form liposomal vesicles. The residual organic solvent was then evaporated at 60 °C for 30 min using a magnetic stirrer at (450 rpm). After cooling the LOS formulations to 25 °C for 30 min, each sample was loaded into a scintillation vial and stored at 4 °C for further evaluation. On the other hand, Vit C solution (10 mg/mL) was prepared by dissolving 10 mg of drug powder into 1 mL of 10 mM succinate buffer solution (pH 3.5) at room temperature.

### 3.3. Physicochemical Characterization of Liposomal Formulations

#### 3.3.1. Morphology of Liposomal Formulations

The morphologies of CL and LOS were observed using transmission electron microscopy (TEM, Tecnai F20 G2; FEI, Hillsboro, OR, USA) [43,44]. Approximately 4 μL of the sample was loaded onto the copper grid and blotted for 1.5 s at 4 °C. Subsequently, the samples were fixed using the plunger freezing method with liquid ethane. The samples were observed at an accelerating voltage of 120 kV.

#### 3.3.2. Vesicular Size and Zeta Potential of Liposomal Formulations

The particle sizes and polydispersity index (PDI) values of the CL and LOSs were determined using a Zetasizer Nano dynamic light scattering particle size analyzer (Malvern Instruments, Worcestershire, UK) [45]. Each sample was added to disposable cells after a 3-fold dilution with distilled water. The zeta potentials of the CL and LOS vesicles were also estimated using a Zetasizer Nano at 25 °C. Samples (100 μL) were loaded into the capillary cell after 3-fold dilution with distilled water, and 20 runs were performed for each measurement. All measurements were carried out in triplicate at 25 °C.

#### 3.3.3. Vit C Content Analysis

The Vit C content of the liposomal formulations was determined by HPLC. Triton X-100 (10%) was added to each liposomal sample, vortexed for 10 min, and centrifuged at 13,000 rpm. The supernatant was diluted 2-fold with 0.1% formic acid. The concentration of Vit C in the sample was analyzed using a Waters HPLC system comprising a pump (Model 515 pump), a UV–VIS (ultraviolet-visible) detector (Model 486), and an autosampler (Model 717 plus). Vitamin C was separated on a C18 column (Kinetex 5 μm EVO C18, 100 Å, 150 × 4.6 mm) under gradient elution conditions using (A) 0.1% formic acid and (B) acetonitrile at a flow rate of 0.8 mL/min [46]. The initial condition was 0% of B, followed by a linear increase of the gradient to 80% of B in 8.5 min, maintaining isocratic conditions for 0.5 min, and finally, a linear decrease to 0% of B from 9 to 20 min. Under this gradient flow, Vit C eluted at approximately 3.4 min of retention time. The column temperature was set to 22 °C, and the samples were cooled at 4 °C in the autosampler. A 30 μL aliquot was injected, and the column eluent was monitored at a wavelength of 245 nm. The least-square linear regression was linear in the Vit C range of 5–100 μg/mL, with a coefficient of determination (*r*^2^) of 0.998.

#### 3.3.4. Loading Efficiency of Vit C in Liposomal Vesicles

To measure the loading efficiency of Vit C-loaded into CL or LOS, an Amicon^®^ stirred cell device (Millipore, Billerica, MA, USA) and 100 kDa pore-sized ultrafiltration discs were used. First, 2–3 mL of the formulation was diluted 10 times with 10 mM succinate buffer (pH 3.0) to obtain 20–30 mL for filtration. After installing the ultrafiltration disk at the bottom of the stirred cell, the diluted solution was filled into the cell, and nitrogen pressure was passed through the cell at a flow rate of 1 L/min. The filtered sample, containing unloaded Vit C molecules, was then diluted 20 times with 0.1% formic acid, and the concentration of Vit C in the samples was quantified using HPLC, as described in Section 3.3.3. After quantifying the amount of unloaded Vit C, the loading efficiency was calculated as following Equation (1) [47].
(1)$$\mathrm{Loading}\;\mathrm{efficiency}\;(\%)=\frac{Initial\;loaded\;amount\;of\;Vit\;C-unencapsulated\;amount\;of\;Vit\;C}{Initial\;loaded\;amount\;of\;Vit\;C}\times 100$$

### 3.4. Photostability of Vit C-Loaded Liposomal Formulations

Approximately 10 mL of Vit C-loaded liposome formulations and drug solution were placed in a transparent 10 mL vial and placed in a xenon lamp chamber (DYX 500A; DY-Tech, Seoul, Republic of Korea) under a light intensity of 20,000 lx at 30.8 ± 0.34 °C. At predetermined time points (8 and 24 h), changes in physical characteristics, including discoloration and phase separation, were visually inspected. In addition, the remaining drug in the samples was analyzed using HPLC [46].

### 3.5. In Vitro Release Profile of Liposomal Formulations

The in vitro release patterns of Vit C from the Vit C solution, Vit C-loaded CL, and LOS were evaluated using the Franz diffusion cell model under light-shielded conditions. Each formulation containing 4 mg of Vit C was added to a regenerated cellulose membrane (SpectraPor^®^ 6; Gyeonggi-do, Republic of Korea, molecular weight cut-off of 8 kDa), and the top of the cell was blocked with Parafilm to prevent foreign substances or evaporation from the loaded formulations. The cells were filled with 10 mM succinate buffer (pH 3.0, 32 °C) and stirred at 450 rpm. Before the test, the membrane was activated with a 10 mM succinate buffer for 24 h. At predetermined times (0.5, 1, 2, 4, 8, 12, and 24 h), 250 μL of dissolution medium was withdrawn and diluted with 0.1% formic acid. The concentration of Vit C in the samples was analyzed using HPLC [46].

### 3.6. Ex Vivo Skin Absorption of Vit C-Loaded Liposomal Formulations

The permeability and the accumulated amount of Vit C from the formulations (Vit C solution, Vit C-loaded CL and LOS) were evaluated using a Franz diffusion cell [48]. Pig dorsal skin (Cronex Co. Ltd., Gyeonggi-do, Republic of Korea), with a thickness of 0.8–1.2 mm, was located between the donor and receptor compartments. In the donor compartment, 0.2 mL of Vit C solution, Vit C-loaded CL and LOS formulations were loaded, and 0.02% sodium thiosulfate containing 10 mM succinate buffer (pH 3.0) was filled in the receptor compartment. Sodium thiosulfate (0.02%) was introduced to prevent the degeneration of the pig dorsal skin, and a succinate buffer (pH 3.0) was used to minimize the degradation of Vit C after diffusing the skin. After loading the samples, the top of the cell was wrapped with Parafilm to prevent contamination and evaporation of the loaded samples. At predetermined time points of 2, 4, 8, 12, and 24 h, 250 μL of receptor solution was withdrawn and diluted 2 times with 0.1% formic acid for HPLC analysis of Vit C. The permeated amount per unit area (mg/cm^2^) versus time (h) was plotted, and permeation parameters such as flux (mg·cm^−2^·h^−1^), lag time (h), and permeability coefficient (10^−6^·cm/h) were calculated as described below. The flux (mg·cm^−2^·h^−1^) was determined from the slope of the linear portion of the permeated amount (mg/cm^2^) versus time (h). The lag time was calculated based on the extrapolation of the linear portion of the permeated amount per unit area (mg/cm^2^) versus the time plot [49]. The permeability coefficient (10^−6^·cm/h) was determined by dividing the flux (mg·cm^−2^·h^−1^) by the drug concentration in the donor phase (mg/cm^3^) [50].

After skin permeability evaluation, both sides of the loaded pig dorsal skin were wiped to remove the agent or solvent. Thereafter, the skins were cut into small pieces, soaked in 10 mL of 10 mM succinate buffer (pH 3.0) containing 0.02% *w/v* sodium thiosulfate and stirred for 24 h. After that, 1 mL of the solution was centrifuged at 13,000 rpm, and the supernatant was diluted with 0.1% formic acid to evaluate the Vit C using HPLC.

### 3.7. Statistical Analysis

Each experiment was performed at least 3 times, and the data are presented as mean ± standard deviation (SD). Statistical significance was determined using a one-way analysis of variance (ANOVA) test and was considered to be significant at *p* < 0.05 unless otherwise indicated. The statistical significance of the permeation parameters obtained with Vit C-loaded CL or LOS was determined using Student’s *t*-test and was considered to be significant at *p* < 0.05.

## 4. Conclusions

A novel multi-layered liposomal system of Vit C was constructed, and its morphology, physicochemical properties, drug stability, and skin permeation profile were successfully evaluated. The incorporation of tricaprylin with phospholipids and Chol increases the hydrophobicity of the liposomal bilayer and contributes to the formation of a multi-layered liposomal structure. The entrapment of the labile compound into the novel system improved the chemical stability of Vit C under light exposure compared to that in CL. Moreover, the multi-layered liposomal system provided comparable *ex vivo* skin permeation and accumulation of Vit C to that of the smaller 200 nm-sized CL. Therefore, this novel liposomal system is expected to be employed for the topical delivery of Vit C with enhanced chemical stability and skin absorption.

## Figures and Tables

**Figure 1 pharmaceuticals-16-00121-f001:**
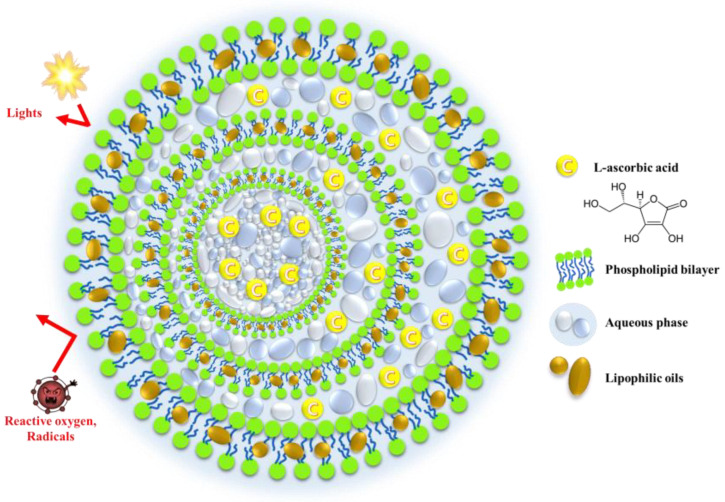
Schematic illustration of the shape of the Vit C-loaded lipo-oil-some (LOS) system and its protective effects. The red arrow signifies the hampering of the penetration of oxygen and light into the liposomal vesicle.

**Figure 2 pharmaceuticals-16-00121-f002:**
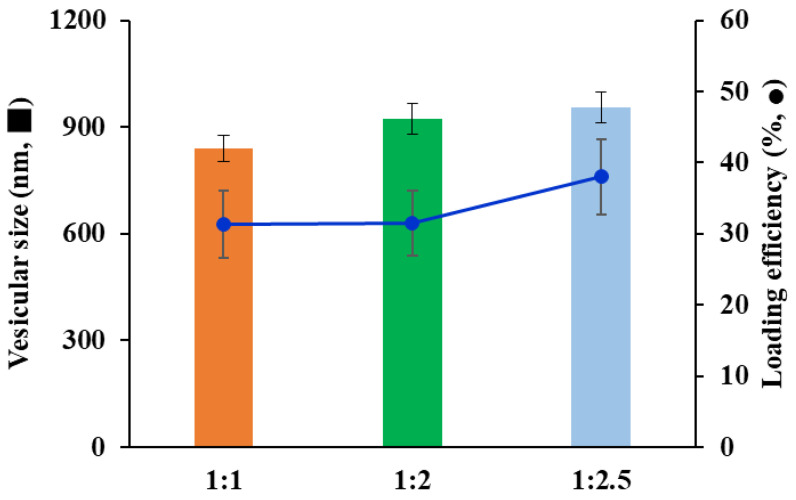
Effect of the oil and water ratio on vesicular size (nm) and loading efficiency (%) of lipo-oil-some (LOS) system. Vesicular size and loading efficiency are expressed as an orange bar and blue circles, respectively. Data represent mean ± SD (n = 3).

**Figure 3 pharmaceuticals-16-00121-f003:**
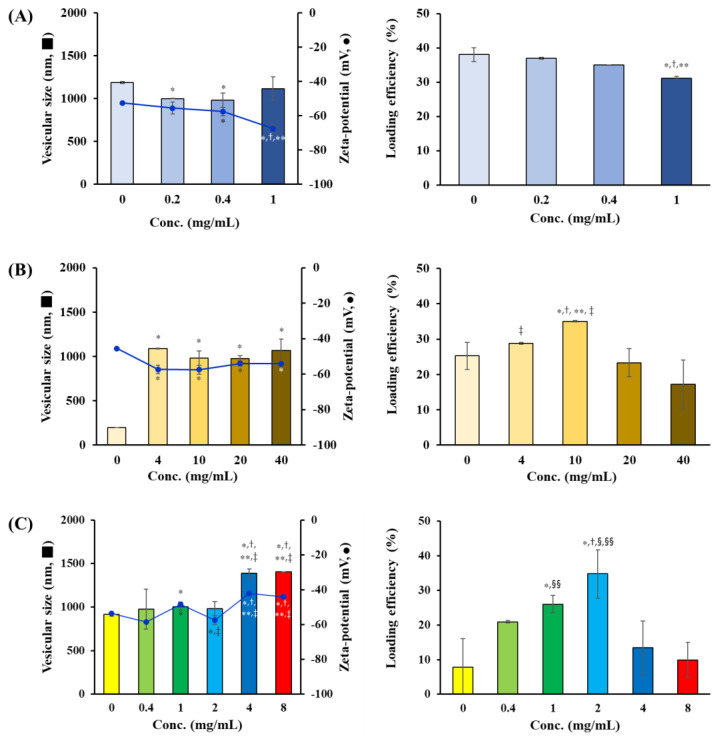
Effect of the amount of (**A**) 1,2-dipalmitoyl-sn-glycerol-3-phosphoglycerol (DPPG), (**B**) tricaprylin, and (**C**) cholesterol on vesicular size and zeta potential of lipo-oil-somes (LOSs) and loading efficiency of Vit C in the LOS system. (**A**) DPPG concentrations of 0, 0.2, 0.4, and 1 mg/mL correspond to LOS1, LOS2, LOS3, and LOS4, respectively. (**B**) Tricaprylin concentrations of 0, 4, 10, 20, and 40 mg/mL correspond to CL, LOS5, LOS6, LOS7, and LOS8, respectively. (**C**) Cholesterol concentrations of 0, 0.4, 1, 2, 4, and 8 mg/mL correspond to LOS9, LOS10, LOS11, LOS12, LOS13, and LOS14, respectively. Data represent mean ± SD (n = 3). Significantly different from LOS1 (* *p* < 0.05), LOS2 (^†^
*p* < 0.05) and LOS3 (** *p* < 0.05) in (**A**); CL (* *p* < 0.05), LOS5 (^†^
*p* < 0.05), LOS7 (** *p* < 0.05), and LOS8 (^‡^
*p* < 0.05) in (**B**); LOS9 (* *p* < 0.05), LOS10 (^†^
*p* < 0.05), LOS11 (** *p* < 0.05), LOS12 (^‡^
*p* < 0.05), LOS13 (^§^
*p* < 0.05), and LOS14 (^§§^
*p* < 0.05) in (**C**) by the ANOVA test.

**Figure 4 pharmaceuticals-16-00121-f004:**
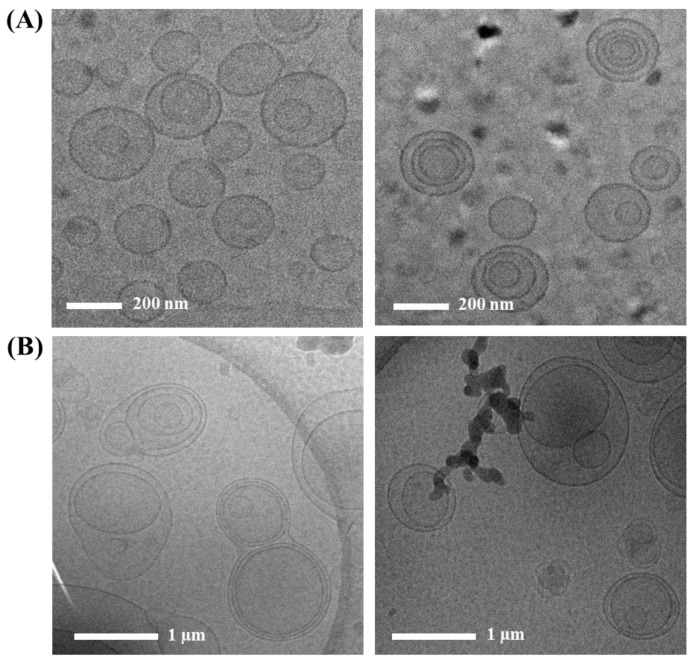
Cryo-transmission electron microscopy (TEM) images of (**A**) Vit C-loaded conventional liposome (CL) and (**B**) Vit C-loaded lipo-oil-some (LOS).

**Figure 5 pharmaceuticals-16-00121-f005:**
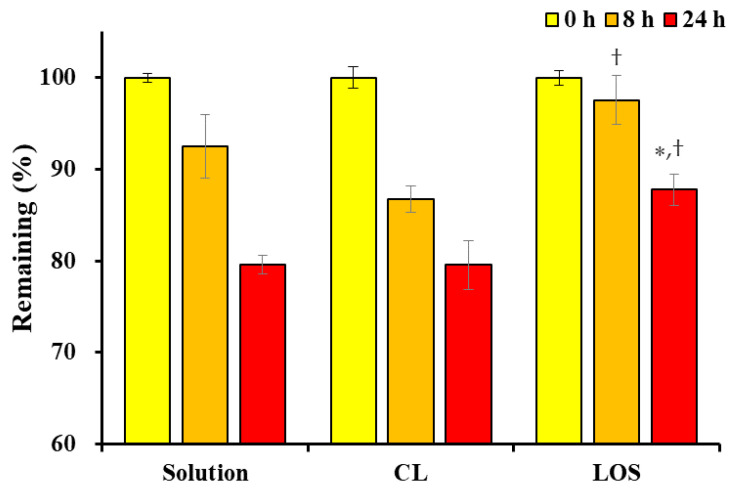
Photostability of Vit C solution, Vit C-loaded conventional liposome (CL) and Vit C-loaded lipo-oil-some (LOS). Data represent mean ± SD (n = 3). Significantly different from Vit C solution (* *p* < 0.05) and CL (^†^
*p* < 0.05) by the ANOVA test.

**Figure 6 pharmaceuticals-16-00121-f006:**
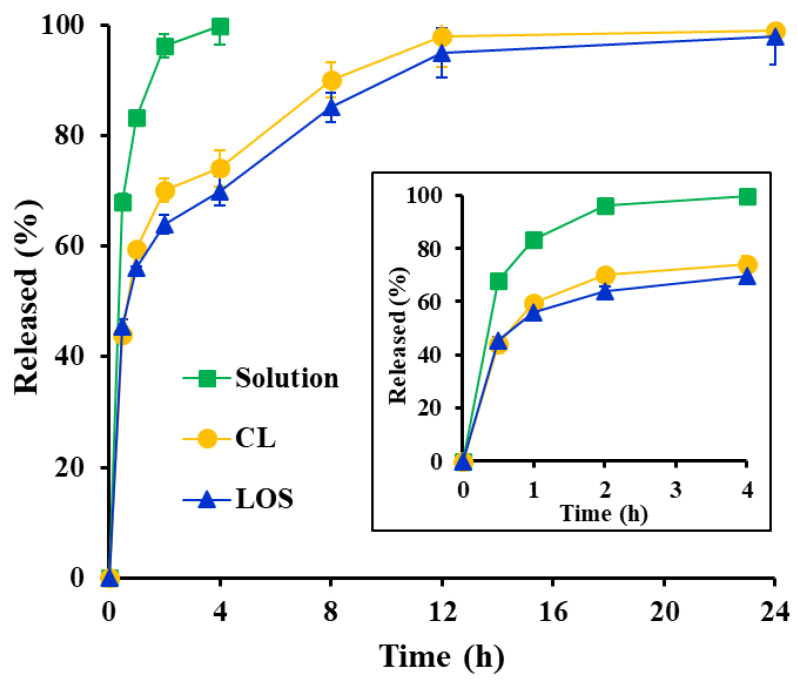
*In vitro* release profile of Vit C from Vit C solution, Vit C-loaded conventional liposome (CL) and Vit C-loaded lipo-oil-some (LOS) in 10 mM succinate buffer (pH 3.0) using a Franz diffusion cell at 32 °C. Data represent mean ± SD (n = 3).

**Figure 7 pharmaceuticals-16-00121-f007:**
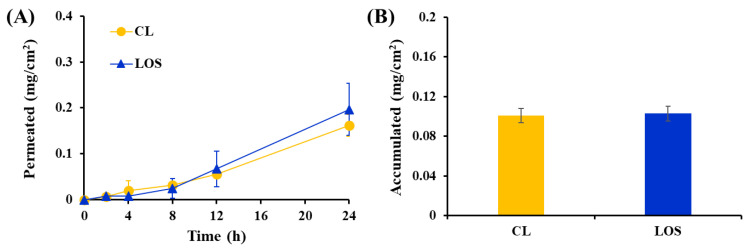
Ex vivo permeation profile of Vit C through pig dorsal skin after topical application of Vit C solution, Vit C-loaded conventional liposome (CL) and Vit C-loaded lipo-oil-some (LOS). (**A**) Permeated amount (mg/cm^2^) and (**B**) *ex vivo* accumulated amount (mg/cm^2^) of Vit C at 24 h post-administration of LOS through pig dorsal skin. Data represent mean ± SD (n = 5).

**Table 1 pharmaceuticals-16-00121-t001:** Compositions of Vit C-loaded conventional liposome (CL) and lipo-oil-somes (LOSs).

Formulation No.	CL	LOS1	LOS2	LOS3	LOS4	LOS5	LOS6	LOS7
Vit C (mg)	10	10	10	10	10	10	10	10
PC (mg)	20	20	20	20	20	20	20	20
DPPG (mg)	0.4	0	0.2	0.4	1	0.4	0.4	0.4
Tricaprylin (mg)	0	10	10	10	10	4	10	20
Cholesterol (mg)	2	2	2	2	2	2	2	2
10 mM succinate Buffer (mL)	q.s.	q.s.	q.s.	q.s.	q.s.	q.s.	q.s.	q.s.
Total (mL)	1.0	1.0	1.0	1.0	1.0	1.0	1.0	1.0
pH ^(a)^	3.5	3.6	3.5	3.5	3.3	3.4	3.4	3.5
**Formulation No.**	**LOS8**	**LOS9**	**LOS10**	**LOS11**	**LOS12**	**LOS13**	**LOS14**	
Vit C (mg)	10	10	10	10	10	10	10	
PC (mg)	20	20	20	20	20	20	20	
DPPG (mg)	0.4	0.4	0.4	0.4	0.4	0.4	0.4	
Tricaprylin (mg)	40	10	10	10	10	10	10	
Cholesterol (mg)	2	0	0.4	1	2	4	8	
10 mM succinate Buffer (mL)	q.s.	q.s.	q.s.	q.s.	q.s.	q.s.	q.s.	
Total (mL)	1.0	1.0	1.0	1.0	1.0	1.0	1.0	
pH ^(a)^	3.4	3.4	3.3	3.5	3.5	3.4	3.4	

^(a)^ pH data are expressed as means (n = 3), and SD values were within 0.1 in all formulations.

**Table 2 pharmaceuticals-16-00121-t002:** *Ex vivo* permeation parameters of Vit C following topical application of Vit C-loaded conventional liposomes (CL) and the optimized lipo-oil-some (LOS12). Data represent mean ± SD (n = 5).

	CL	LOS
Permeated (mg/cm^2^)	0.162 ± 0.023	0.197 ± 0.056
Flux (mg·cm^−2^·h^−1^)	0.007 ± 0.001	0.008 ± 0.002
Lag time (h)	1.455 ± 0.731	2.097 ± 0.841
Permeability coefficient (10^−6^·cm/h)	0.718 ± 0.078	0.909 ± 0.235

## Data Availability

Not applicable.

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
