# Peer review of "Design of Novel Tricaprylin-Incorporated Multi-Layered Liposomal System for Skin Delivery of Ascorbic Acid with Improved Chemical Stability"

_pharmaceuticals, 2023, doi:10.3390/ph16010121_

Round 1

Reviewer 1 Report

The manuscript entitled “Design of novel tricaprylin-incorporated multi-layered liposomal system for skin delivery of ascorbic acid with improved chemical stability” shows interesting results and informative knowledge for researchers in the field of pharmaceutics.  The authors proposed the novel formulations of a delivery system of ascorbic acid to overcome its poor stability called lipo-oil-somes (LOSs). However, there are some comments and questions for this manuscript as follows:

1. The authors proposed the formulations of LOSs, which can be accepted as a novel system for transdermal delivery of ascorbic acid. Thus, this manuscript is attractive for the researchers in a field of drug delivery. 

2. The picture of LOSs containing ascorbic acid shown in Fig 1. indicates that ascorbic acid was located in aqueous phase of the liposomes with some tricaprylin molecules, which is a medium chain triglyceride. 

Due to the lipophilicity of tricaprylin, why did the authors suppose that tricaprylin would be located in the same phase as ascorbic acid?   

3. The authors should show the evidences from the determination of locations of tricaprylin and ascorbic acid in the liposomes in the manuscript.

4. The authors should determine the solubility of ascorbic acid in tricaprylin and show the results in the manuscript.

5. The authors proposed that the more DPPG in the formulations led to the lower loading efficiency of ascorbic acid in LOSs because of the repulsion between the carboxyl group in ascorbic acid molecules and negative charges on the liposomal surface (P.5). 

Could the authors check again for the sources of the negative charges in the ascorbic acid molecules?

6. The authors should indicate the statistical difference (p-values) between the data from different test samples that are shown in Fig 3, 5, 7 (b).

7. The authors should propose the mechanisms of LOSs formation and explain why this system provided multilayered-LOSs instead of emulsions. This information should be also added into the manuscript.

Reviewer 2 Report

Myoung Jin Ho et al., submitted a manuscript entitled Design of a novel tricaprylin-incorporated multi-layered liposomal system for skin delivery of ascorbic acid with improved chemical stability. The paper was well written, yet there are a few things that can be modified 

1. Introduction - The introduction should end with the objectives of the current research work

2. Figure 5. Mention what is mean by SOLUTION.

3. Ex-vivo studies need to perform to confirm the transdermal delivery.

Round 2

Reviewer 1 Report

The authors responded to the reviewers' comments and revised the manuscript following the suggestions. This manuscript can be thus accepted in the present form.